# Predicting nonsense-mediated mRNA decay from splicing events in sepsis using RNA-sequencing data

Jaewook Shin[1] , Alger M Fredericks[1], Brandon E Armstead[1], Alfred Ayala[1], Maya Cohen[2], William G Fairbrother[3], Mitchell M Levy[2], Kwesi K Lillard[1], Emanuele Raggi[1], Gerard J Nau[4], Sean F Monaghan[1,5]

**Alternative splicing (AS) and nonsense-mediated mRNA decay (NMD) are highly conserved cellular mechanisms that modulate gene expression. Here, we introduce the NMD pipeline that computes how splicing events introduce premature termination codons to mRNA transcripts via frameshift, then predicts the rate of premature termination codon–dependent NMD. We use whole-blood, deep RNA-sequencing data from critically ill patients to study gene expression in sepsis. Statistical significance was determined as adjusted $P < 0.05$ and $|\log_2$ fold change$| > 2$ for differential gene expression and probability ≥0.9 and $|DeltaPsi| > 0.1$ for AS. The NMD pipeline was developed based on the AS data from Whippet. We demonstrate that the rate of NMD is higher in the sepsis and deceased groups compared with the control and survived groups, which may signify aberrant splicing because of altered physiology in critical illness. Predominance of non-exon skipping events was associated with disease and mortality states. The NMD pipeline also revealed proteins with potential association with sepsis. Together, these results emphasize the utility of the NMD pipeline in studying AS-NMD along with differential gene expression analysis and uncovering proteins associated with sepsis.**

## Introduction

Alternative splicing (AS) and nonsense-mediated mRNA decay (NMD) are crucial molecular processes that modulate gene expression (1, 2). AS contributes to protein diversity in higher eukaryotes, and close to 95% of multiexon genes are estimated to undergo AS in major human tissues (3). NMD is a highly conserved surveillance mechanism that eliminates mRNAs based on multiple criteria, most commonly when premature termination codons (PTCs) are located more than 50–55 nucleotides upstream of the final exon junction (4). Because splicing events can generate PTCs via frameshift, AS-NMD may down-regulate genes or decay aberrant transcripts (5), thereby fine-tuning gene expression and maintaining cellular homeostasis (6). Recently, AS and NMD have been indicated in the pathogenesis of malignancy (7, 8, 9), in critical illness (10, 11, 12), and in the changes seen in gene expression because of altered physiological states such as hypoxia, acidosis, and temperature (13, 14).

Sepsis is a leading cause of mortality worldwide responsible for up to 1 in 5 deaths (15), and occurs when infection causes a dysregulated host response leading to life-threatening organ dysfunction (16). Although earlier diagnosis and guidelines have shown some benefit, current understanding of sepsis pathogenesis has not substantially improved patient outcomes (17). Despite the efforts to understand sepsis-induced cellular and subcellular dysfunction by characterizing gene expression profiles (18, 19, 20, 21), none have translated to bedside treatments (22). Given that differential gene expression (DGE) alone has been ineffective, studying AS-NMD in conjunction can delineate how intermediate steps of gene expression affect downstream proteins.

Here, we demonstrate a computational pipeline that predicts NMD from AS data generated from patients' whole-blood, deep RNA-sequencing (RNA-Seq) data. This pipeline computes how splicing events generate PTCs via frameshift and affect protein levels and how such prediction identifies proteins with potential association in sepsis and mortality. For this work, we focus on PTC-dependent NMD that targets PTCs located 50–55 bp upstream of the final exon junction. We hypothesize that there will be more splicing events leading to NMD in the sepsis and deceased group because of altered physiology that either purposefully or aberrantly decays select transcripts.

## Results

### Patient characteristics of control versus sepsis groups and survived versus deceased groups

A total of 43 critically ill patients with sepsis and six without sepsis were studied. Among sepsis patients, the mortality rate was 34.9%

[1]Division of Surgical Research, Department of Surgery, Rhode Island Hospital/Alpert Medical School of Brown University, Providence, RI, USA    [2]Division of Pulmonary, Critical Care, and Sleep Medicine, Rhode Island Hospital/Alpert Medical School of Brown University, Providence, RI, USA    [3]Department of Molecular Biology, Cell Biology, and Biochemistry, Brown University, Providence, RI, USA    [4]Division of Infectious Diseases, Department of Medicine, Rhode Island Hospital/Alpert Medical School of Brown University, Providence, RI, USA    [5]Division of Trauma and Critical Care, Department of Surgery, Rhode Island Hospital/Alpert Medical School of Brown University, Providence, RI, USA

Correspondence: sean_monaghan@brown.edu

**Life Science Alliance**

with 25 patients who survived and 18 who deceased. The control group was younger than the sepsis group (43.8 ± 18 yr versus 63.2 ± 14.9 yr, $P$ = 0.049) but did not have significantly different percentages of males (33.3% versus 55.8%, $P$ = 0.55) or non-Caucasians (50% versus 27.9%, $P$ = 0.53). Similarly, patients' age, sex, and race did not differ based on mortality. Compared with controls, the sepsis group had statistically significantly higher rate of shock (0% versus 53.5%, $P$ = 0.02), longer median ICU length of stay (LOS) (1.3 versus 2.2 d, $P$ < 0.01), and longer median hospital LOS (2.8 versus 9.2 d, $P$ < 0.01). Survived and deceased groups had a similar rate of shock (56% versus 50%, $P$ = 0.76), SOFA score (4.5 versus 6, $P$ = 0.29), ICU (2 versus 4.4 d, $P$ = 0.05), and hospital LOS (9.5 versus 7.8 d, $P$ = 0.61) (Table 1).

### Whole-blood, non-poly(A) selected, deep RNA-Seq pipeline enables DGE and AS studies

Next, deep RNA-Seq of patients' whole blood yielded a total of 11.1 billion reads across all samples. Non-poly(A) tail selection facilitated the comprehensive AS analysis because all transcripts with and without poly(A) tail were included. Of the total reads, 8.8 billion reads were mapped to the human genome with 2.3 billion reads unmapped (Fig 1A). Mean total RNA-Seq reads per sample were similar between control and sepsis groups (120 versus 116 million, $P$ = 0.57) and survived and deceased groups (118 versus 112 million, $P$ = 0.14), confirming the efficacy of deep RNA sequencing yielding at least 100 million reads per sample (Table 1). The mapped reads were then processed for downstream analyses including DGE analysis of 17,043 genes and AS studies with 220,779 splicing events (Fig 1A).

### More up-regulated genes but a similar rate of AS in sepsis compared with controls

First, we examined the DGE and AS profiles of control and sepsis groups. Of the 17,043 genes analyzed in the two groups, 1,349 genes (7.9%) were significantly differentially expressed with 1,325 up-regulated (98.2%) and 24 (1.8%) down-regulated in sepsis showing that more genes analyzed were highly expressed in sepsis (Figs 1B and S1). There were 220,779 splicing events analyzed in control versus sepsis, with 2,158 splicing events (1%) significantly differentially frequent. Of these, 1,014 events (47%) were more frequent in sepsis and 1,144 events (53%) less frequent (Figs 1C and S2). Of note, the median percent spliced in (psi) value in Whippet—the proportion of reads that include splicing events in the final transcript sequence across samples—of the control group was 1.98% and of the sepsis group was 40.4% ($P$ < 0.0001), demonstrating that the sepsis group had a statistically higher magnitude of splicing events compared with the control group. We then categorized 220,779 splicing events into subtypes so that the splicing events mediated exclusively by the splicing machinery could be distinguished from alternative transcription sites. These subtypes were then compared between all splicing events (effectively representing the "control" group) and splicing events statistically more or less frequent in sepsis ("sepsis" group). The results showed that the alternative transcription events represented 48.1% and 55% of all the splicing events in each group, with

transcription start (TS) and end (TE) constituting the highest percentages (94.5%, 95.6%), whereas splicing events were 51.9% and 45% in each group with exon skipping events (ES) constituting the highest percentages (76.3%, 44.7%) (Fig 1D, Table S1). Then, we selected four splicing events—exon skipping (ES), retained intron (RI), alternative acceptor (AA), and alternative donor (AD)—and compared their frequency between the control and sepsis groups, showing that ES was significantly less frequent in sepsis (76.3% versus 44.7%, $P$ < 0.001), whereas other events were significantly more frequent in sepsis (RI 9.5% versus 19.5%, $P$ < 0.001; AA 8.1% versus 18%, $P$ < 0.001; AD 6.1% versus 17.8%, $P$ < 0.001), demonstrating that predilection for non-ES splicing events was present in sepsis (Fig 1E, Table S2).

### More down-regulated genes but a similar rate of AS in the deceased compared with the survived group

Next, we examined the DGE and AS profiles of sepsis patients based on their mortality status. Of the 16,837 genes analyzed in survived and deceased groups, 118 genes (0.7%) were significantly differentially expressed with 7 up-regulated (5.9%) and 111 (94.1%) down-regulated in sepsis showing that more genes analyzed were significantly less expressed in the deceased group (Figs 1F and S3). There were 233,753 splicing events analyzed in the two groups with 2,282 significantly more or less frequent (1%). Of these, 1,172 events (47%) were more frequent in the deceased group and 1,110 events (53%) less frequent, again highlighting the similar degree of splicing in contrast to the DGE results (Figs 1G and S4). We then categorized 233,753 splicing events into splicing and alternative transcription subtypes. The results were consistent with control versus sepsis analysis, showing that the alternative transcription events represented 49.5% and 62.1% in each group, with TS and TE constituting the highest percentages (94.6%, 97%), whereas splicing events were 50.5% and 37.9% in each group with ES being the highest percentages (76.1%, 48.5%) (Fig 1H, Table S3). The frequency of all the splicing events ("survived" group) and splicing events statistically more or less frequent ("deceased" group) showed that ES was significantly less frequent in deceased (76.1% versus 48.5%, $P$ < 0.001), whereas all other events were significantly more frequent in deceased (RI 9.5% versus 21.7%, $P$ < 0.001; AA 8.2% versus 15.6%, $P$ < 0.001; AD 6.2% versus 14.2%, $P$ < 0.001), demonstrating a similar pattern between sepsis and deceased groups regarding splicing events (Fig 1I, Table S4).

### Developing a computational pipeline to predict NMD with splicing data from Whippet

To study NMD, we reasoned that the AS data from Whippet (23) can show if and how many PTCs would be generated from each splicing event. The scope of the pipeline was to predict PTC generation by four splicing events (ES, RI, AA, and AD) based on the established principle that the presence of PTC is expected to elicit NMD (13). Thus, we selected key splicing information, such as Ensembl ID, splicing event type, splicing coordinate, node, and strand to model the nucleotide sequences of mature transcripts resulting from splicing events and to identify how frameshifts generate PTCs upstream of 50–55 base pairs from the final exon junction in

**Table 1. Patient demographics, clinical outcomes, and RNA-Seq data in control versus sepsis and survived versus deceased.**

| | Control | Sepsis | *P*-value | Survived | Deceased | *P*-value |
|---|---|---|---|---|---|---|
| Sample size (N) | 6 | 43 | — | 25 | 18 | — |
| Age, mean (years) | 43.8 ± 18.4 | 63.2 ± 14.9 | 0.049 | 63.9 ± 16.3 | 61.9 ± 12.2 | 0.66 |
| Male (%) | 2 (33.3%) | 24 (55.8%) | 0.55 | 13 (46.4%) | 11 (73.3%) | 0.17 |
| Non-Caucasian (%) | 3 (50%) | 12 (27.9%) | 0.53 | 8 (28.6%) | 4 (26.7%) | 0.84 |
| Mortality (%) | 0 (0%) | 15 (34.9%) | 0.16 | — | — | — |
| Shock (%) | 0 (0%) | 23 (53.5%) | 0.02 | 14 (56%) | 9 (50%) | 0.76 |
| SOFA (score) | 4 | 5.5 | 0.06 | 4.5 | 6 | 0.29 |
| ICU stay, median (days) | 1.3 | 2.2 | <0.01 | 2 | 4.4 | 0.05 |
| Hospital stay, median (days) | 2.8 | 9.2 | <0.01 | 9.5 | 7.8 | 0.61 |
| Total reads, mean (N) | 120,199,186 | 116,110,683 | 0.57 | 118,008,653 | 112,567,805 | 0.14 |
| Mapped, mean, N (%) | 96,239,315 (80.1%) | 92,755,097 (79.9%) | 0.47 | 92,177,756 (78.2%) | 93,832,802 (83.4%) | 0.63 |
| Unmapped, median, N (%) | 21,895,386 (18.2%) | 19,715,777 (17%) | 0.99 | 24,944,197 (21.8%) | 17,869,942 (16.6%) | <0.01 |

Mortality refers to the rate of in-hospital death. SOFA score refers to the Sequential Organ Failure Assessment (SOFA) score. Total reads refer to the total number of RNA-Seq reads yielded from each patient group, with respective percentages of reads mapped to the human genome ("mapped") and not mapped to the human genome ("unmapped").

accordance with an accepted prerequisite for NMD ([4]). As a result, our NMD pipeline yields the following outputs: the predicted frame of each transcript, the number and location of all PTCs generated per frame, and predicted NMD true or false based on the predicted frame and PTC location (Fig 2A).

### Higher rate of predicted NMD in the sepsis group from non-ES splicing events

Using the NMD pipeline, we compared the overall rate of NMD in sepsis compared with the control. We processed all splicing events and those significant in sepsis through the pipeline and found a significantly higher rate of NMD predicted to occur from splicing events in sepsis compared with control (90.7% versus 93.3%, *P* = 0.03), and each splicing subtype predicted to induce NMD at a similar rate, except for AD (94.4% versus 89.2%, *P* = 0.03) (Fig 2B, Table S5). ES was the most common splicing event predicted to cause NMD though less frequently in sepsis (80.5% versus 39.4%, *P* < 0.001). There were significantly higher percentages of RI, AA, and AD splicing events predicted to induce NMD in sepsis (RI 10.7% versus 27%, *P* < 0.001; AA 4.3% versus 16.4%, *P* < 0.001; AD 4.4% versus 17.2%, *P* < 0.001) (Fig 2C, Table S6). Although ES generated the highest median number of PTCs, none of the splicing events in sepsis introduced significantly different amounts of PTCs compared with all splicing events (ES 37 versus 36, *P* = 0.51; RI 23 versus 21, *P* = 0.68; AA 24 versus 24, *P* = 0.74; AD 30 versus 35, *P* = 0.43), showing that the number of PTCs generated is not correlated with a higher rate of NMD seen in sepsis (Fig 2D, Table S7).

### Higher rate of predicted NMD in the deceased group from non-ES splicing events

We also studied whether the rate of NMD would be significantly different based on mortality and found a significantly higher rate of NMD predicted to occur in the deceased group (90.8% versus

93.3%, *P* = 0.04) and each splicing subtype predicted to induce NMD at a similar rate (Fig 2E, Table S8). Similarly, ES most predicted NMD though less frequently in the deceased group (80.5% versus 40%, *P* < 0.001). Again, there were significantly higher percentages of RI, AA, and AD events predicted to induce NMD in the deceased group (RI 10.7% versus 30.9%, *P* < 0.001; AA 4.3% versus 13.9%, *P* < 0.001; AD 4.5% versus 15.2%, *P* < 0.001) (Fig 2F, Table S9). ES accounted for the highest median number of PTCs generated, but splicing events did not introduce meaningfully different median numbers of PTCs, consistent with control versus sepsis analysis that the number of PTCs generated does not correlate with the higher rate of NMD in the deceased group (ES 37 versus 39.5, *P* = 0.73; RI 23 versus 19, *P* = 0.07; AA 24 versus 18, *P* = 0.47; AD 31 versus 28.5, *P* = 0.58) (Fig 2G, Table S10).

### NMD pipeline can identify proteins with potential novel roles in sepsis and mortality

We then examined whether the splicing events not predicted to induce NMD, thus preserving certain transcripts, could identify proteins with potential association with sepsis and mortality. The rationale was that the genes not predicted to undergo NMD based on our pipeline may be important to study in sepsis because they are part of the minority of genes less likely to be decayed. We performed GO enrichment analysis of all splicing events yielding predicted NMD result as "false" (NMD-F) that are then filtered by *P* < 0.01. In control versus sepsis, 45 splicing events were NMD-F and 13 genes were identified as highly likely to be relevant to seven biological processes, including essential nucleic acid metabolism (e.g., DMP, tRNA), cell division and development (e.g., meiosis, epithelium), inflammation (e.g., *NF-kB*, phosphorylation), and response to stressor (e.g., lead) (Fig 2H, Table S11). In survived versus deceased, 39 splicing events were NMD-F and 11 genes were identified as highly likely to be relevant to six biological processes, including essential nucleic acid metabolism (e.g., NAD), cell

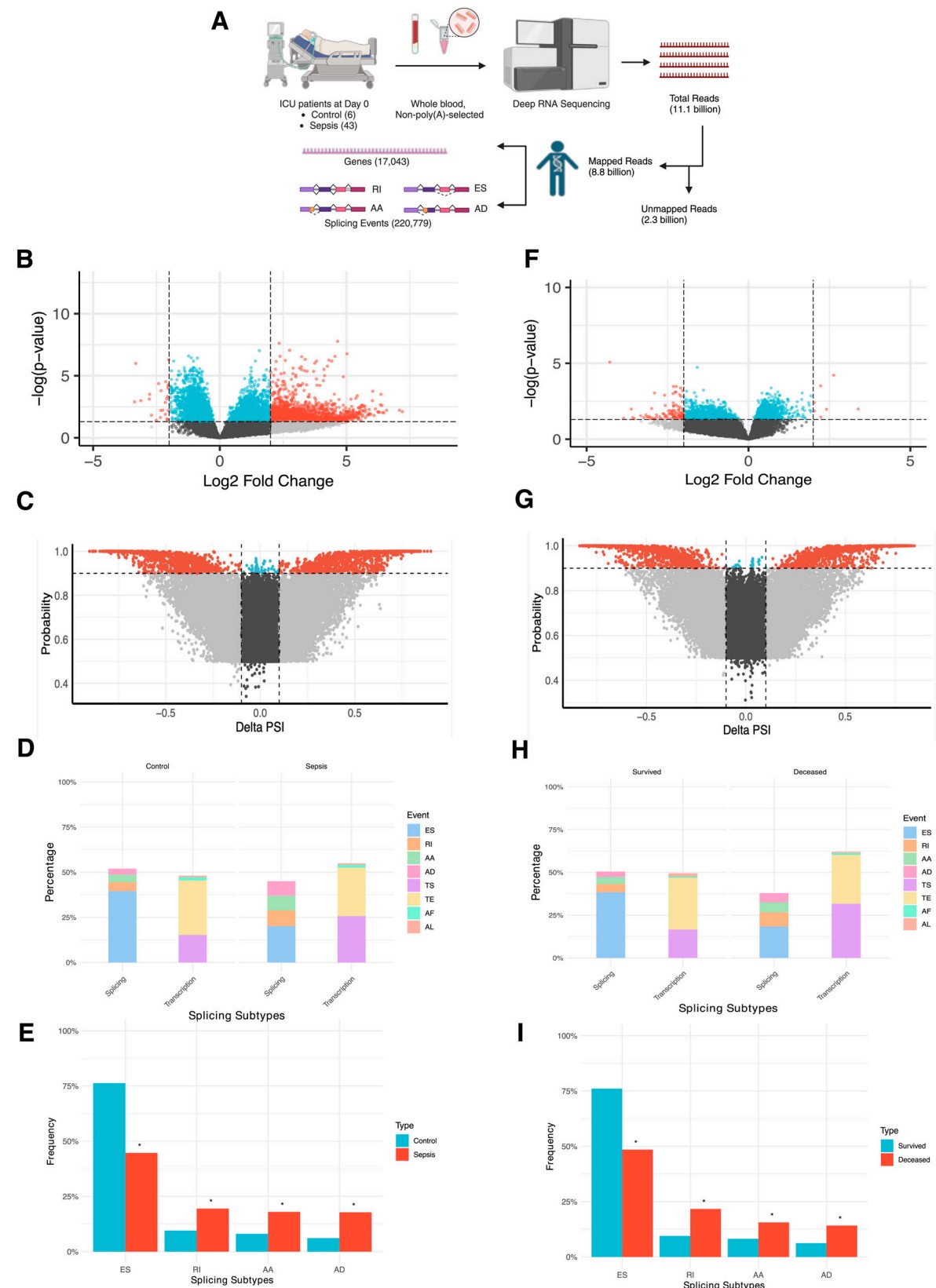

**Figure 1. Differential gene expression (DGE) and alternative splicing (AS) data in control versus sepsis and survived versus deceased groups.**
**(A)** Diagram describing RNA-Seq workflow from ICU patients to their DGE and AS information. Created in https://BioRender.com. **(B)** Volcano plot showing DGE analysis of significantly up- or down-regulated genes in control versus sepsis based on adjusted *P*-value and log$_2$ fold change (log$_2$FC) (red), adjusted *P*-value alone (blue), log$_2$FC alone (gray), and not statistically significant results (black). The x-axis (log$_2$FC) represents log$_2$ of sepsis/control. **(C)** Volcano plot showing differential splicing analysis of

division and development (e.g., cell cycle, *CDK*), inflammation (e.g., phosphatidylcholine), and response to stressor (e.g., alcohol, UV) (Fig 2I, Table S12). Thus, NMD-F from the NMD pipeline can identify proteins with essential cellular functions such as inflammation and nucleic acid and cell metabolism.

### Plasma grancalcin concentration is higher in sepsis as predicted by the NMD pipeline

Finally, we used proteomics to test the NMD pipeline predictions and evaluate the impact of NMD on protein abundance. *Grancalcin* (*GCA*) was chosen because it was the only gene with significantly more or less frequent splicing events in both sepsis and deceased groups and with one of the highest RNA-Seq read counts overall in both groups (Fig 3A). Between control and sepsis, *GCA* was not significantly differentially expressed based on DGE, less frequently spliced in sepsis based on Whippet, and NMD predicted true based on the NMD pipeline. Thus, a higher protein level was predicted in sepsis because of fewer splicing events inducing NMD (Fig 3B). Based on the splicing coordinate, we found that the AD event would occur in intron 1 and the density map of the GCA genome confirmed that there were fewer reads in sepsis (3,868 reads) compared with control (4,202 reads) in that coordinate, suggesting fewer AD events occurring in sepsis (Fig 3C). ELISA results showed that the median plasma GCA concentration was statistically significantly higher in the sepsis group compared with controls (0 ng/ml versus 0.81 ng/ml, $P$ = 0.04), consistent with the NMD pipeline prediction (Fig 3D). Weak positive correlation between the RNA-Seq read counts and protein concentrations in sepsis (R = 0.19, $P$ = 0.29) and moderate negative correlation in control (R = −0.41, $P$ = 0.42) were not statistically significant (Fig 3E). Plasma granulin did not yield significant results in control versus sepsis (Figs S5 and S6).

### Plasma GCA concentration is higher in deceased as predicted by the NMD pipeline

Between the survived and deceased group, *GCA* was also not significantly different based on DGE, less frequently spliced in sepsis based on Whippet, and NMD predicted true based on the pipeline. Thus, a higher protein level was predicted in the deceased group (Fig 3F). Based on the splicing coordinate, we found that an ES event would occur in exon 2 and the density map of the *GCA* genome confirmed that there were higher reads in deceased (10,887 reads) compared with survived (3,601 reads) in that coordinate, supporting that there were fewer ES events in the deceased group, and thus more reads given higher inclusion of exons

(Fig 3G). ELISA results showed that the median plasma GCA concentration was statistically significantly higher in the deceased group compared with survived (0.79 ng/ml versus 0.89 ng/ml, $P$ = 0.006), in concordance with the NMD pipeline prediction (Fig 3H). Although not statistically significant, higher RNA-Seq read counts demonstrated a trend toward moderate positive correlation with protein concentrations in the deceased group (R = 0.52, $P$ = 0.08), whereas it was not correlated in control (R = 0.04, $P$ = 0.85) (Fig 3I).

## Discussion

We have developed a pipeline to predict how splicing events introduce PTCs into mRNA transcripts and induce PTC-dependent NMD. We showed that although more genes were up-regulated in sepsis and down-regulated in deceased based on DGE, a significantly higher proportion of transcripts were predicted to undergo NMD from splicing events in sepsis and deceased, which can influence downstream protein levels. In addition, we have shown that the NMD-F group can identify protein targets with potential association in sepsis and that proteomics results on plasma GCA were consistent with the NMD pipeline prediction.

Our RNA-Seq data and downstream analyses have high biological and clinical relevance because they originated from the whole blood of 49 critically ill patients in the ICU with and without sepsis. This enabled an unbiased study of each patient's transcriptome, augmented by the depth of 100 million reads that allowed a comprehensive DGE analysis and non-poly(A) tail selection that incorporated splicing intermediates to facilitate AS studies. Although DGE data showed up-regulation of genes in sepsis and down-regulation of genes in deceased patients, AS data showed transcripts were differentially spliced in nearly equivalent levels in both groups. Given AS has been suggested to maintain cellular homeostasis in diseased states (24), the distribution of splicing data unique from DGE indicates the potential importance of including AS in gene expression studies. Unlike ES—the most common form of splicing events (25)—non-ES events (RI, AA, AD) were more common in sepsis and mortality, which indicates that non-ES events may exert a greater effect on sepsis and mortality status. To mitigate the loss of variance from the relative nature of RNA-Seq, we used ARC, quality control, and high-sequencing depth. A significant difference between up- and down-regulated DGE results may be attributed to a higher $\log_2$FC threshold we used.

Based on the healthy human and mouse cohort splicing data, some studies have established that at least 33% of the cassette exons are frame-preserving (2, 26), which may indicate that NMD is

---

significantly more or less frequent splicing events in control versus sepsis based on probability and delta percent spliced in (DeltaPsi) (red), probability alone (blue), DeltaPsi alone (gray), and not statistically significant results (black). The x-axis (DeltaPsi) represents the percentage of splicing in sepsis subtracted by the percentage of splicing in control. **(D)** Proportion of each subtype out of all splicing events in control versus sepsis groups that are then categorized into "splicing" and "transcription-related" groups. AF refers to alternative first exon, and AL refers to alternative last exon. **(E)** Frequency of each of the four splicing events (from the "splicing" group in Fig 3D) in percentage in control versus sepsis. **(B, F)** Volcano plot showing DGE analysis of significantly up- or down-regulated genes in survived versus deceased with same color and statistical depictions as (B). The x-axis ($\log_2$FC) represents $\log_2$ of deceased/survived. **(C, G)** Volcano plot showing differential splicing analysis of significantly more or less frequent splicing events in survived versus deceased with same color and statistical depiction as (C). The x-axis (DeltaPsi) represents the percentage of splicing in deceased subtracted by the percentage of splicing in survived. **(H)** Proportion of each subtype out of all splicing events in survived versus deceased groups that are then categorized into "splicing" and "transcription-related" groups. (I) Frequency of each of the four splicing events (from the "splicing" group in Fig 3H) in percentage in survived versus deceased.

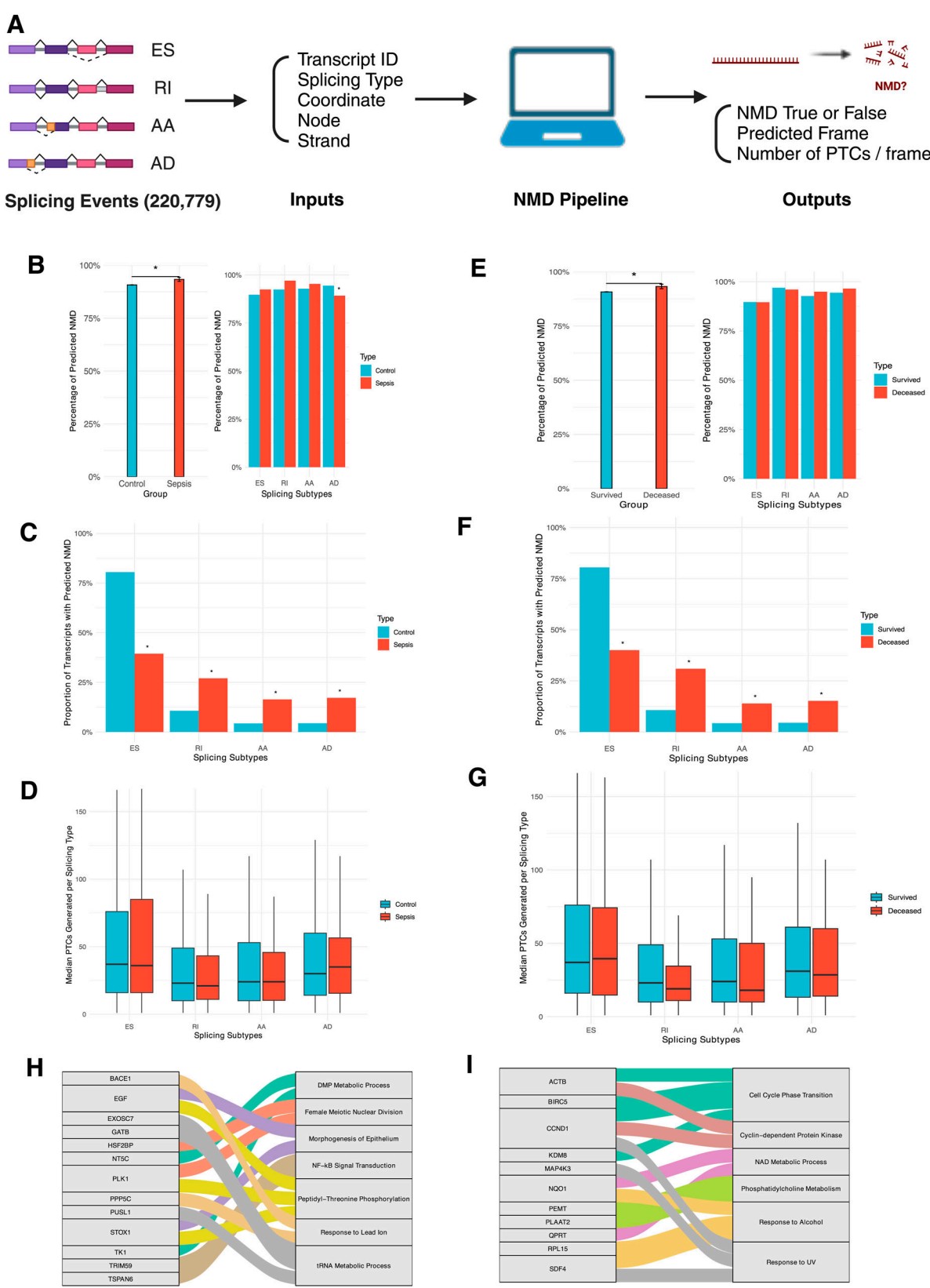

**Figure 2. Alternative splicing (AS) and nonsense-mediated mRNA decay (NMD) data in control versus sepsis (Fig 1B–D and H) and survived versus deceased groups (Fig 1E–G and I).**
**(A)** Diagram describing the development of the NMD pipeline from Whippet AS data to NMD outputs. Created in https://BioRender.com. **(B)** Bar graph showing the percentage of splicing events predicted to induce NMD in control versus sepsis (left) and the percentage of predicted NMD stratified by splicing subtypes (right). **(C)**

predicted to occur in up to 66% of exon skipping events. Although these data provide helpful references, our study used a fundamentally different methodology to investigate splicing events. Although the two prior studies used human and mouse transcripts from Alternative Splicing Database and lymphoblastoid cell lines derived from 40 to 86 Yoruba individuals to yield exon skipping results, our study used prospectively obtained whole-blood samples from critically ill patients with and without sepsis, then performed RNA sequencing, and analyzed the raw data with Whippet to yield splicing event data in critically ill patients. In addition, the major hypothesis of our work is that altered physiology of critical illness as seen in sepsis is associated with changes in alternative splicing, as our previous work has shown that critical illness affects splicing events to modify downstream protein translation, specifically how physiologic derangement leads to a change in exon skipping events that render transmembrane proteins to become soluble (10, 11). Given sepsis is marked by physiologic derangement, it is not inconsistent, but expected that the exon skipping results from our cohort of critically ill patients had different exon skipping frequency compared with noncritically ill humans, mice, and cell lines from the two studies. Although causality cannot be established with the current data, our results can supplement prior studies by reporting a higher percentage of exon skipping events predicted to induce NMD in the critically ill control group, which indicates critical illness can affect splicing percentages.

More than 90% rate of NMD seen across all groups can support that NMD is a conserved, ubiquitous subcellular process. A higher rate of NMD in sepsis and mortality is likely explained by acidosis, hypoxia, and fever or hypothermia that predispose cells to aberrant splicing (10, 11, 12) and splicing errors that necessitate a higher rate of NMD (2). Moreover, the AS-NMD mechanism may also be down-regulating certain transcripts related to cellular stress signals (4) or affected by hypoxia (13) to regulate resources devoted to combating infection and preventing organ failure. Given the known coupling of AS-NMD in down-regulating certain transcripts (5), a list of genes identified as NMD-F (Fig 2H and I) have noteworthy involvement in nucleic acid and cell metabolism, signal transduction, inflammation, and response to stressor as evidenced by the GO enrichment analysis. Of note, all but 1 gene (*PLAAT2*) included in the NMD-F group had statistically nonsignificant DGE results, and thus were analyzed in GO terms as aggregates. Thus, the NMD pipeline not only highlights the AS-NMD interaction, but also identifies some potential proteins with roles in inflammation and essential cellular metabolism. Although NMD changes are not targetable as therapeutics given its high conservation and ubiquitous functions in high eukaryotes, the AS-NMD dynamic in critical illness can illuminate previously unknown pathophysiology and proteins potentially associated

with sepsis. The fact that the number of PTCs was not significantly different in control versus sepsis and survived versus deceased is consistent with the literature that shows NMD machinery is also sensitive to the location of PTC, such as downstream of the last exon junction complex, near the stop codon, and long exons with far distances between the PTC and the stop codon (27).

It is important to consider the intermediate transcriptional regulatory steps in the gene expression pathway that affect protein levels such as the AS-NMD mechanism. To this end, we have shown that although DGE of *GCA* in the sepsis and deceased groups was not significantly different, the NMD pipeline predicted higher GCA levels in both conditions, which was corroborated by the plasma protein ELISA. We selected grancalcin (GCA) as our main protein target because it was one of the most abundant targets found based on the RNA-Seq read counts and it was the only target with significantly different splicing events in both control versus sepsis and survived versus deceased groups. Plasma samples were used to test the targets given their availability in our laboratory and their practicality as a potential clinical test. For this reason, we also selected plasma granulin (GRN) as an additional protein target to test for control versus sepsis.

Given that RNA-Seq read counts and protein concentration data were not significantly correlated (Fig 3E and I), other decay pathways (28, 29) and post-translational modifications may have affected the protein level, which may explain why plasma granulin was not consistent with the NMD pipeline prediction (Figs S5 and S6). In addition, a low detection rate in the control group may have been due to a lower proportion of GCA present in the plasma samples from less severe inflammatory conditions compared with the sepsis group. Regardless, the NMD pipeline can potentially help uncover proteins previously unknown because of DGE studies alone. Because the AS-NMD mechanism is highly conserved and essential throughout high eukaryotes (3), other fields can also use the NMD pipeline.

Some limitations include the potential confounding effect of the age difference between the control and sepsis groups and non-NMD decay processes that could have influenced the transcript and protein levels. In addition, an increase in NMD-substrate levels may also be attributed to a decrease in NMD activity—thus leading to an increase in NMD targets—which the scope of our study does not include but can be a plausible mechanism. Our study does not establish definitive causality between AS-NMD machinery and gene expression in sepsis. For instance, the AS and NMD interplay in *GCA* has not been validated experimentally on a molecular level because of limitations in obtaining further blood samples and availability in laboratory resources. However, we do provide a potential mechanistic insight of how altered splicing events can lead to

Proportion of splicing events of transcripts predicted to cause NMD per splicing subtype in control versus sepsis. **(D)** Box plot showing the median number of premature termination codons (PTCs) generated per splicing subtype in control versus sepsis. **(E)** Bar graph showing the percentage of splicing events predicted to induce NMD in survived versus deceased (left) and the percentage of predicted NMD stratified by splicing subtypes (right). **(F)** Proportion of splicing events of transcripts predicted to cause NMD per splicing subtype in survived versus deceased. **(G)** Box plot showing the median number of premature termination codons (PTCs) generated per splicing subtype in survived versus deceased. **(H)** Sankey diagram showing all the genes with $P < 0.01$ in GO enrichment analysis and their respective biological processes in control versus sepsis. **(I)** Sankey diagram showing all the genes with $P < 0.01$ in GO enrichment analysis and their respective biological processes in survived versus deceased.

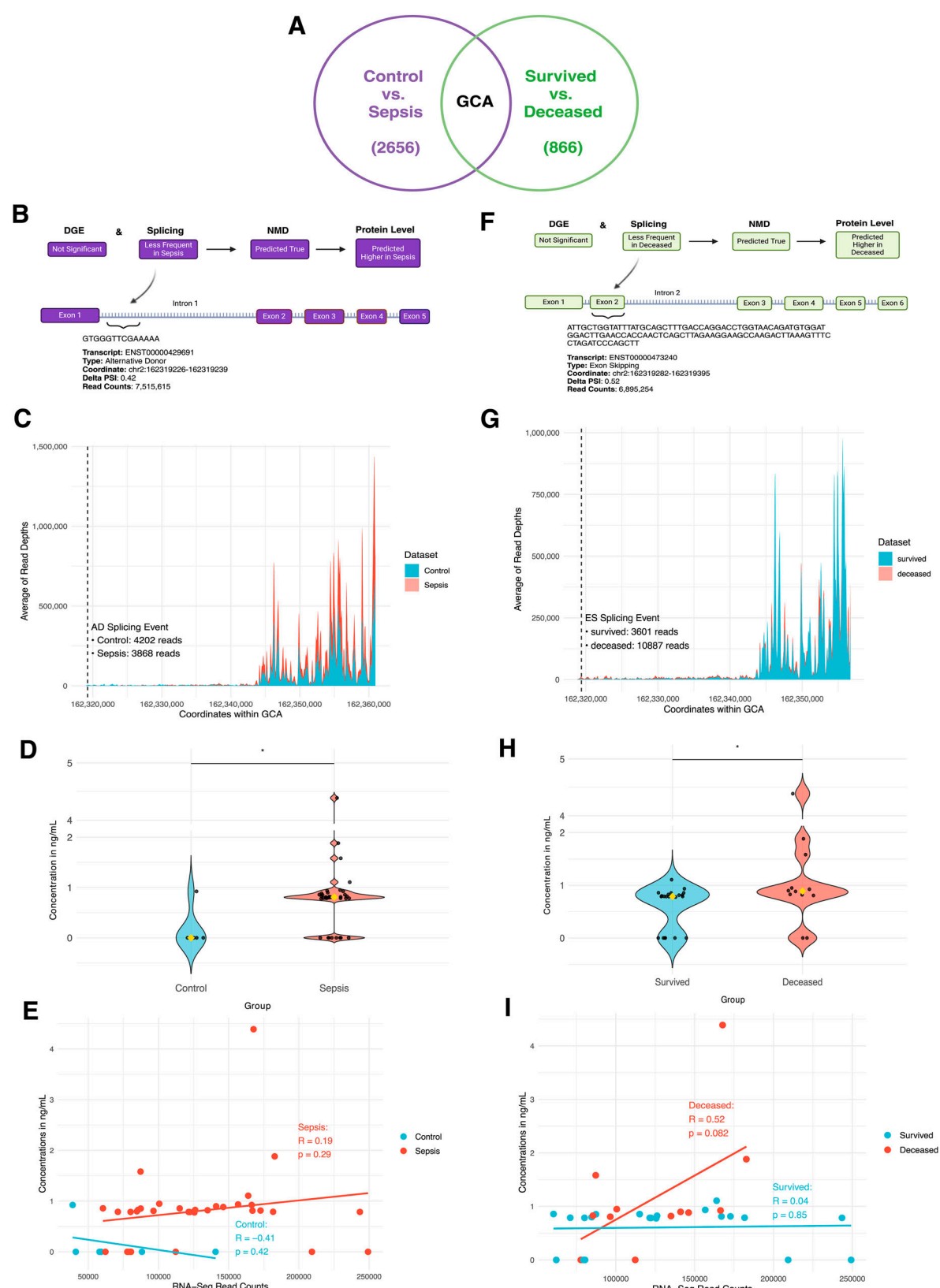

**Figure 3. NMD pipeline prediction and proteomics data on plasma grancalcin (GCA) in control versus sepsis and survived versus deceased groups.**
**(A)** Diagram describing GCA as the only gene with significant differential splicing in both control versus sepsis (total 2,656 significant differential splicing events) and survived versus deceased (total 866 significant differential splicing events) and with one of the highest absolute RNA-Seq read counts (ARC). Created in https://BioRender.com. **(B)** Diagram showing prediction on plasma GCA protein level based on differential gene expression, splicing, and NMD data, along with the details on its

 **Life Science Alliance**

NMD in sepsis via a novel computational pipeline developed based on the evidence-based guidelines. First, our NMD pipeline was built to use splicing event outputs from Whippet software as inputs to predict the rate of NMD. Whippet (23) operates on a peer-reviewed and established computational pipeline that yields splicing events from RNA-Seq data. Second, we used the statistical significance threshold recommended by Whippet documentation (30) to analyze only the splicing events more likely to be significantly different between the control and sepsis groups. Third, the coding algorithm in our pipeline was written based on one of the canonical ways that the NMD machinery is triggered—the introduction of a premature stop codon from a frameshift splicing event at 50-55 base pairs upstream of the final exon junction (4). Thus, despite the above limitations, our results have undergone validations with ELISA and have adhered to evidence-based, validated methods from input selection to coding implementation. Future studies can include an external validation of our computational work with a separate cohort of critically ill patients or using alternative software that extracts splicing isoform sequences to find sequence commonality in a gene potentially significant in sepsis. Although the control group had a relatively small sample size compared with the sepsis group, from our sample size of around 50 patients, the depth of our RNA-Seq data covering at least 100 million RNA reads per patient provided sufficient data points for our current investigation of predicting the rate of NMD from splicing events.

Overall, this study demonstrates that the NMD pipeline can predict NMD from splicing events. Although NMD changes alone are not causative of sepsis or directly treatable as targets, our pipeline enables the study of AS-NMD and their potential interplay in sepsis and builds upon the previous literature on the role of splicing in critical illness. In the sepsis and deceased groups, we report higher rates of NMD, either from aberrant splicing requiring more frequent NMD or from purposeful down-regulation of certain genes. We also propose that the NMD pipeline can be a component of gene expression studies, both in sepsis and in other fields, to characterize post-transcriptomic gene products at levels detected under homeostasis and pathological conditions. Studying AS-NMD in conjunction with DGE studies can capture the nuances of the complex gene expression pathway in sepsis, for which DGE analysis alone may not suffice. Furthermore, investigating the role of AS-NMD in altered physiological states can enhance current understanding of the role of splicing in critical illness and uncover potential proteins associated with sepsis.

## Materials and Methods

### Study approval

All patients or their appropriate surrogates provided informed consent as approved by the hospital Institutional Review Board (Approval #: 411616).

### Study design

A single-center, prospective study of critically ill patients with and without sepsis at an academic tertiary center was performed from 2021 to 2022.

Inclusion criteria for a diagnosis of sepsis were as follows: corresponding ICD-10 coding on admission, confirmed diagnosis of sepsis by attending physician, documentation of septic shock with evidence of hypoperfusion (mean arterial pressure < 65, systolic blood pressure < 90 mmHg, lactate > 2 mmol/liter after 30 ml/kg crystalloid bolus within 3 h of identification), and the evidence of infection with at least one end-organ failure per SOFA score. Exclusion criteria included age less than 18 yr, pregnancy, incarceration, trauma within 30 d, immunocompromised conditions such as malignancy and pulmonary fibrosis, and known recent antibiotic use within 1 wk before admission (11). Patients were enrolled from the medical intensive care unit upon admission or within 24 h of developing sepsis in the ICU.

Whole-blood samples were drawn in PAXgene tubes (QIAGEN) on hospital day 0, then sent to Genewiz for RNA extraction, globin and ribosomal RNA depletion, and deep RNA sequencing on Illumina HiSeq machines. Polyadenylic acid (poly(A)) tail selection was not performed to include all splicing intermediates and events. The sequencing provided 150-bp paired-end reads and at least 100 million reads per sample (31).

The raw sequencing data were assessed for quality control with FastQC (32), then aligned to the most recent assembly (GCF_000001405.40) of Genome Reference Consortium Human Build 38 (GRCh38) (33) with the STAR aligner (34). Reads that aligned to GRCh38 ("mapped" reads) were separated from "unmapped" reads. The mapped reads from the control versus sepsis and survived versus deceased group then underwent analysis for DGE and AS.

For DGE, featureCounts (35) was used to yield raw absolute read counts (ARC) so that the loss of variance from normalization can be mitigated; then, the DESeq2 package from Bioconductor (36) helped identify differentially expressed genes. Statistical significance was determined as adjusted $P < 0.05$ and $|\log_2$ fold change$| > 2$ (12). For AS, Whippet (23) was used to compare exon skipping (ES),

---

significant differential splicing event in control versus sepsis. Created in https://BioRender.com. **(C)** Density map showing the average number of RNA-Seq reads per 150-bp coordinate range of the GCA genome. **(B)** Dashed line represents the location of the alternative donor (AD) event (from (B)) and respective number of reads in control versus sepsis at the coordinates of the AD event. **(D)** Violin plot showing the distribution and median ELISA protein concentrations of GCA in each sample in control versus sepsis. **(E)** Graph showing the correlation data between ELISA concentrations in ng/ml and RNA-Seq read counts of GCA in control versus sepsis. **(F)** Diagram showing prediction on plasma GCA protein level based on differential gene expression, splicing, and NMD data, along with the details on its significant differential splicing event in survived versus deceased. Created in https://BioRender.com. **(G)** Density map showing the average number of RNA-Seq reads per 150-bp coordinate range of the GCA genome. **(F)** Dashed line represents the location of the exon skipping (ES) event (from (F)) and respective number of reads in survived versus deceased at the coordinates of the ES event. **(H)** Violin plot showing the distribution and median ELISA protein concentrations of GCA in each sample in survived versus deceased. **(I)** Graph showing the correlation data between ELISA concentrations in ng/ml and RNA-Seq read counts of GCA in survived versus deceased.

retained intron (RI), alternative donor (AD), and alternative acceptor (AA) events. Of note, Whippet output includes core exon (CE) events, which refer to exons involved in exon skipping; thus, we used the nomenclature ES for clearer delineation. Statistical significance was determined by probability ≥ 0.9 and |DeltaPsi| > 0.1 per Whippet documentation (30), which represent the magnitude of splicing event difference between groups. Probability is a Bayesian estimate of a given splicing event differentially spliced, and DeltaPsi is the percent difference in splicing events between two groups. We performed an additional analysis to calculate the magnitude of splicing events with appropriate contextualization using the Anderson–Darling normality test, identifying the median of the absolute values of individual DeltaPsi in control and sepsis groups, then using the Wilcoxon rank-sum test.

For the NMD pipeline, we leveraged Whippet's AS output data to develop a computational approach to examine the rate of NMD, number of PTCs generated, stratification by splicing subtype, and GO enrichment analysis on splicing events with predicted false NMD. Specifically, we wrote a code in R script that uses splicing event information such as Ensembl gene id, a splicing subtype, a splicing event coordinate, and positive or negative strand of the transcript with the splicing event to predict whether each splicing event would generate PTCs, thereby predicting to induce NMD. Thus, the outputs of our code included the predicted frame of codons of each transcript based on the UCSC genome browser, the number of PTCs to be introduced by each splicing event for each possible frame, and a true versus false result of whether NMD would be induced based on whether PTCs would be generated in a predicted frame. Of note, the scope of our code was to predict PTC generation that elicits PTC-dependent NMD for PTCs 50–55 bp upstream of the final exon junction. Also, our NMD pipeline was designed to process all the splicing events designated as ENSEMBL canonical transcripts because they are the most representative transcripts that balance the highest coverage of conserved exons, expression, and consistency with other resources such as NCBI, and thus was chosen as an input to our NMD pipeline (Tables S5 and S10).

We performed enzyme-linked immunosorbent assay (ELISA) of grancalcin (GCA) (Catalog #: MBS2709681; MyBioSource) and granulin (GRN) (Catalog #: EH367RB; Invitrogen) with adherence to commercially available ELISA protocols from respective manuals. Two replicates of each sample were used to calculate an average of the two optical density (OD) values to compute final concentrations. Concentrations of any OD below the detectable range indicated by the manufacturer were imputed as zero.

All computational and statistical analyses were done in R (37) and command line *bash*, *awk*, *grep* script. For continuous variables, either the *t* test or Wilcoxon test was used based on the Shapiro–Wilk normality test. For categorical variables, a chi-square test with or without Yates' correction for continuity was used based on the sample sizes.

## Data Availability

Our datasets and code used and developed for the computational pipeline are available in Zenodo: DGE, AS, and NMD datasets (38) and computational pipeline for NMD (39).

## Supplementary Information

## Acknowledgements

We thank R Zhao of the Division of Surgical Research in the Department of Surgery at Rhode Island Hospital for valued instructions with proteomics work; and BioRender for allowing the creation of several subfigure flowcharts. This work was supported by Armand D Versaci Research Scholar in Surgical Sciences Award (to J Shin), National Institutes of Health grant P20GM121344 (to AM Fredericks and GJ Nau), National Institutes of Health grant T32GM065085-20 (to BE Armstead), National Institutes of Health grant R35GM118097-09 (to A Ayala), National Institutes of Health grant T32HL134625 (to M Cohen), National Institutes of Health grant R01GM127472-06 (to WG Fairbrother), National Institutes of Health grant R01HL162954-03 (to MM Levy), National Institutes of Health grant T32HL134625-08 (to KK Lillard), and National Institutes of Health grant R35GM142638-04 (to SF Monaghan).

### Author Contributions

J Shin: conceptualization, software, formal analysis, validation, investigation, visualization, methodology, and writing—original draft, review, and editing.
AM Fredericks: conceptualization, data curation, software, formal analysis, supervision, investigation, methodology, and project administration.
BE Armstead: supervision, methodology, and writing—review and editing.
A Ayala: resources, supervision, project administration, and writing—review and editing.
M Cohen: resources and funding acquisition.
WG Fairbrother: supervision, methodology, and writing—review and editing.
MM Levy: resources and funding acquisition.
KK Lillard: data curation, software, visualization, and writing—review and editing.
E Raggi: software, methodology, and writing—review and editing.
GJ Nau: resources, supervision, funding acquisition, and writing—review and editing.
SF Monaghan: conceptualization, resources, data curation, supervision, funding acquisition, validation, investigation, methodology, project administration, and writing—review and editing.

### Conflict of Interest Statement

The authors declare that they have no conflict of interest.

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
