## [Reviewer comments · Life Science Alliance]

Predicting Nonsense-mediated mRNA Decay from Splicing Events in Sepsis using RNA-Sequencing Data

Jaewook Shin, Alger Fredericks, Brandon Armstead, Alfred Ayala, Maya Cohen, William Fairbrother, Mitchell Levy, Kwesi Lillard, Emanuele Raggi, Gerard Nau, and Sean Monaghan

DOI: <https://doi.org/10.26508/lsa.202503380>

Corresponding author(s): Sean Monaghan, Rhode Island Hospital

Review Timeline:

Submission Date:	2025-05-06
Editorial Decision:	2025-07-09
Revision Received:	2025-08-06
Editorial Decision:	2025-08-25
Revision Received:	2025-09-09
Accepted:	2025-09-15

Scientific Editor: Tim Fessenden

Transaction Report:

July 9, 2025

Re: Life Science Alliance manuscript #LSA-2025-03380-T

Sean F Monaghan
Rhode Island Hospital
Surgery

Dear Dr. Monaghan,

Thank you for submitting your manuscript entitled "Predicting Nonsense-mediated mRNA Decay from Splicing Events in Sepsis using RNA-Sequencing Data". The manuscript has been evaluated by expert reviewers, whose reports are appended below. Unfortunately, after an assessment of the reviewer feedback, our editorial decision is against publication in Life Science Alliance at this time.

As you will see, reviewers diverged in their overall appraisal of this work, although some concerns were shared among them. In particular Reviewers 2 and 3 remarked on the limitations of this study due to sample size and cohort demographics, raised concerns over the accuracy of the NMD prediction pipeline, and critiqued claims based on correlative data. Reviewer 2 also asked for the overall magnitude of alternative splicing events, noted potential discrepancies related to skipped exon events, and requested improvements to the text. We concur with this reviewer that the value of Figures 2H and 2I is not clear.

Although your manuscript is intriguing, I feel that the points raised by the reviewers are more substantial than can be addressed in a typical revision period. If you wish to expedite publication of the current data, it may be best to pursue publication at another journal.

Given our interest in the topic, I would be open to re-submission to Life Science Alliance of a significantly revised and extended manuscript that fully addresses all the reviewers' concerns and is subject to further peer review. If you would like to resubmit this work to Life Science Alliance, you may submit an appeal directly through our manuscript submission system. An appeal must contain a revision plan with a point-by-point rebuttal to the reviewer comments. Please note that priority and novelty would be reassessed at re-submission.

Regardless of how you choose to proceed, we hope that the comments below will prove constructive as your work progresses.

Thank you for thinking of Life Science Alliance as an appropriate place to publish your work.

Sincerely,

Reviewer #1 (Comments to the Authors (Required)):

Minor revision:

The authors introduce NMD pipeline to provide a potential mechanistic insight of how altered splicing events can lead to NMD in sepsis via a novel computational pipeline. Based on the RNA-Seq data of whole blood from 49 critically ill patients in the ICU and downstream analysis, it's demonstrated that the rate of NMD is higher in sepsis and deceased groups compared to control and survived groups, which signify purposeful downregulation of transcripts by AS-NMD or aberrant splicing due to altered physiology. And Grancalcin (GCA) was used to test the NMD pipeline to predict and evaluate the effect of NMD on protein abundance. NMD pipeline can potentially help discover protein targets previously unknown due to DGE studies alone. The NMD pipeline not only help in understanding the pathophysiological processes of sepsis but also offer new directions for its diagnosis and treatment. The manuscript can be improved in some aspects before accepted:

- (1) Alternative splicing (AS) introduced PTC into RNA. The data analysis results showed that the sepsis and deceased group had a higher NMD rate. However, compared with the control group, there was no significant difference in the number of PTC. This seems to be contradictory to the significant difference in the NMD rate. Maybe it be caused by the difference in the location of PTC, please provide a reasonable explanation for this.
- (2) Some genes of the sepsis and deceased group were less or more frequently spliced. Whether the sequences of these genes can be analyzed to find the potential specificity or commonality among them.

Reviewer #2 (Comments to the Authors (Required)):

In this manuscript, the authors analyze RNA-Seq data from sepsis and control human patients to determine whether altered RNA processing under these conditions generates putative Nonsense-Mediated Decay (NMD) substrates. While there are some potentially interesting observations-particularly regarding grancalcin-the connection to NMD remains unclear.

Overall I believe that there are interesting findings in their data but that the authors have to seriously reconsider how they analyze it and how they present their results. In light of this, I am not in favor of publication.

Major Concerns:

1- Clarity and Writing Quality:

The manuscript is poorly written and often difficult to understand. It contains excessive abbreviations-some of which are never defined (e.g., AF and AL in Figure 1D, H)-and many sections are nearly incomprehensible due to ambiguous phrasing and insufficient context (see "minor concerns" below for some examples).

2- Patient Group Composition:

The patient groups differ substantially in age (43 vs. 63 years) and gender (33% vs. 55% male) which may confound the results obtained.

3- Magnitude of Alternative Splicing Events:

The manuscript never addresses the extent to which alternative RNA processing events differ between groups. It is possible that a particular exon is skipped 1% vs 5% of the time in sepsis vs control patients. This "statistically different" event is interesting, but very different in nature from the situation where a particular exon is skipped 1% vs. 95% in the two populations. We are never given a sense whether these "statistically different" events are small or large in magnitude.

4- Interpretation of NMD-Linked Events:

The claim that exon skipping promotes NMD in >66% of cases implies that skipped exons are not frame-preserving at a higher rate than chance. This is contrary to established literature. Prior studies (e.g., PMID: 17719121, 39223315) show that at least 33% of cassette exons are frame-preserving. This goes up in conserved skipped exons. Thus AT MOST we expect that exon skipping should produce an NMD substrate 66% of the time. The analysis appears inconsistent with this and raises concerns about the validity of the NMD predictions. Similar issues exist for other types of alternative splicing documented by the authors.

5- Speculative Interpretation of NMD Rates:

In the discussion, the authors suggest that increased NMD in sepsis is due to aberrant splicing, which is highly plausible. However, as discussed above, the observed NMD levels exceed those expected from random or non-functional RNA generation. This discrepancy is never addressed and again suggests a possible flaw in the NMD prediction pipeline.

Minor Comments:

1- Figure 1B-C lacks clear labeling. It's unclear whether the X-axis represents sepsis/control ratios or the reverse.

2- The way the text is written is sometimes ambiguous and hard to parse. Here are several examples:

Line 71. "The sepsis group was older than the controls (vs, $p=0.049$)" - logically the numbers should follow the order of the text and should be "63.2{plus minus}14.9 years vs, 43.8{plus minus}18 years". I know that we can infer that the opposite order is implied, but this cannot be done for the following statements (e.g., which group, sepsis or control, has 33% males?).

Line 91 "Of the 17,043 genes differentially expressed between the two groups ..." This is not correct and very confusing. It should read "Of the 17,043 genes analyzed in the two groups ..."

Line 94 "There were 220,779 events differentially spliced in control versus sepsis ..." I presume that you mean that there were 220,779 RNA processing/splicing events analyzed in control and sepsis samples, since only 1% were significantly different. The rest of the paragraph, and the entire next section, were extremely hard to parse. When the author express a fact, such as "alternative transcription events represented 48.1%" it is imperative that they express what they exactly is the reference point (i.e. 48.1% of what group?).

3- The section starting on page 170 analyzes putative functional consequences of alterations in RNA processing that result in non-NMD substrates. The paucity of hits (13 and 11 genes) makes this unlikely and Figure 2H-1 is meaningless. The authors claim "the NMD pipeline can identify proteins with known sepsis-relevant or essential cellular functions ... not previously known in sepsis" But the changes observed may not be relevant to sepsis and may simply be serendipitous. Leaving aside the paucity of affected genes, how do the authors know that these are relevant to sepsis? Correlation does not mean causation. This is akin to finding that one difference between parked and moving cars is that the former have parking tickets and concluding that the parking tickets caused the cars to become parked.

4- Line 187 "Grancalcin (GCA) was chosen since it was the only gene with significant differential splicing events in both sepsis and deceased groups, each with 2,656 and 866 events," I do not know what these event counts mean. Are these individual reads? How do these events in the aggregate differ between the groups?

5- Figure 3I, the two trends are identical except for a few outliers. This should be eliminated.

Reviewer #3 (Comments to the Authors (Required)):

My comments are as follows that I would like the authors to address:

1. The study claimed about the development of the pipeline/method for detecting and assessing nonsense-mediated mRNA decay (NMD) but it is unclear if validation of the performance was performed. To be clear the method is not detailed in the main text. There might be something in the supplemental doc that I missed
2. It is unclear if detection of AS and NMD of genes such as GCA were experimentally validated.
3. The control group seems to have small sample size (n=6) in comparison.
4. 'While DGE data showed upregulation of genes in sepsis and downregulation of genes in deceased patients, AS data showed transcripts were differentially spliced in nearly equivalent levels in both groups. This suggests AS may modulate downstream effects of over- or under-translation of proteins to maintain cellular homeostasis in diseased states (24). ' Its unclear as written/data illustrated that it has that implication.

Overall, the work is novel and impressive.

Editorial

As you will see, reviewers diverged in their overall appraisal of this work, although some concerns were shared among them. In particular Reviewers 2 and 3 remarked on the limitations of this study due to sample size and cohort demographics, raised concerns over the accuracy of the NMD prediction pipeline, and critiqued claims based on correlative data. Reviewer 2 also asked for the overall magnitude of alternative splicing events, noted potential discrepancies related to skipped exon events, and requested improvements to the text. We concur with this reviewer that the value of Figures 2H and 2I is not clear.

We thank you for the opportunity and continued support of our manuscript. We have a few comments and questions regarding some of the points raised by the reviewers and the Editorial Board.

Regarding further experimental validations of our NMD pipeline, we have validated our computational work with the ELISA validation of the protein Grancalcin (GCA). This work utilized clinical samples paired with samples taken from patients for RNA sequencing. We are not able to obtain more blood from these patients as some patients were recruited as early as 2021. In addition, we do not think animal model experiments are appropriate for sepsis work and this is consistent with NIGMS (funding organization of the R35) policies specific to sepsis (<https://www.nigms.nih.gov/sites/nigms/files/migrated/nagmsc-working-group-sepsis-report.pdf>) with current plans to expand this across the entire NIH. Our lab has focused on human samples, RNA sequencing, and computational methods with focused experimental validations such as ELISA as stated earlier and we do not believe an animal model would translate this work. Thus, we acknowledge our limitations and will edit the manuscript if acceptable to remove speculative language.

Regarding a concern, “insufficient evidence to validate that the NMD pipeline used in this study is correctly identifying splicing events,” we would like to clarify the following points. First, our NMD pipeline is built to use splicing event outputs from Whippet software as inputs to predict the rate of NMD. Whippet (1) operates on a peer-reviewed and established computational pipeline that yields splicing events from RNA-Seq data. In addition, we use statistical significance threshold recommended by Whippet documentation (2) to analyze only the splicing events more likely to be significantly different between control and sepsis groups. Third, coding algorithm in our pipeline was written based on one of the conventional ways that NMD machinery is triggered – introduction of premature stop codon from a frameshift splicing event at 50-55 base pairs upstream of final exon junction (3). Thus, we believe that our pipeline is correctly identifying splicing events since it has adhered to evidence-based, validated methods from input selection to coding implementation.

Regarding a concern, “... given the discrepancy in the measured NMD transcripts from skipped exon events,” we would like to emphasize that our methodology and the two papers cited by Reviewer 2 are fundamentally different, thus the results will imply discrepant findings. In addition, the major hypothesis of our work is that altered physiology of critical illness as seen in sepsis is associated with changes in alternative RNA splicing. The two papers utilized human and mouse transcripts from Alternative Splicing Database (4) and lymphoblastoid cell lines derived from 40-86 Yoruba individuals (5) to yield their exon

skipping results. We have prospectively obtained whole blood samples from critically ill patients with and without sepsis, perform RNA sequencing, then analyzed the raw data with Whippet to yield splicing events data. Our previous work has shown that critical illness affects splicing events to modify downstream protein translation (6, 7), specifically how physiologic derangement leads to a change in exon skipping events that render transmembrane proteins to become soluble. Given sepsis is marked by physiologic derangement, it is not inconsistent, but actually expected, that the exon skipping results from our cohort of critically ill patients had different exon skipping frequency compared to non-critically ill humans, mice, and cell lines from the two studies. Although causality can't be established with the current data, there is work showing clinically relevant physiologic derangements (pH, temperature, hypoxia) impact alternative splicing.

We genuinely appreciate your comments and support. We are open to any other feedback to improve our manuscript for publication in Life Science Alliance. Please let us know if there are any clarifications to be made or other recommended ways to validate our results or changes in the manuscript that could allow for publication.

1. Sterne-Weiler T, Weatheritt RJ, Best AJ, Ha KCH, and Blencowe BJ. Efficient and Accurate Quantitative Profiling of Alternative Splicing Patterns of Any Complexity on a Laptop. *Mol Cell*. 2018;72(1):187-200 e6.
2. T. Sterne-Weiler NA, G. Massman. Whippet. <https://github.com/timbitz/Whippet.jl>.
3. Lareau LF, Brooks AN, Soergel DA, Meng Q, and Brenner SE. The coupling of alternative splicing and nonsense-mediated mRNA decay. *Adv Exp Med Biol*. 2007;623:190-211.
4. Zhang C, Krainer AR, and Zhang MQ. Evolutionary impact of limited splicing fidelity in mammalian genes. *Trends Genet*. 2007;23(10):484-8.
5. Fair B, Buen Abad Najjar CF, Zhao J, Lozano S, Reilly A, Mossian G, et al. Global impact of unproductive splicing on human gene expression. *Nature Genetics*. 2024;56(9):1851-61.
6. Monaghan SF, Chung CS, Chen Y, Lomas-Neira J, Fairbrother WG, Heffernan DS, et al. Soluble programmed cell death receptor-1 (sPD-1): a potential biomarker with anti-inflammatory properties in human and experimental acute respiratory distress syndrome (ARDS). *J Transl Med*. 2016;14(1):312.
7. Monaghan SF, Banerjee D, Chung CS, Lomas-Neira J, Cygan KJ, Rhine CL, et al. Changes in the process of alternative RNA splicing results in soluble B and T lymphocyte attenuator with biological and clinical implications in critical illness. *Mol Med*. 2018;24(1):32.

Another point that may bear some further consideration is point 3 by Reviewer 2. We encourage you to include the magnitude of splicing events with appropriate contextualization of these as the result of a probabilistic analysis.

Thank you for the comment. We performed additional analysis to calculate the magnitude of splicing events with appropriate contextualization using Anderson-Darling normality test, identifying the median of the absolute values of individual delta psi in control and sepsis groups, then using Wilcoxon rank-sum test. Median splicing percentages of control group were 1.98% and of sepsis group were 40.4% (p < 0.0001), demonstrating that sepsis group had statistically higher magnitude of splicing events compared to control group.

Reviewer 1

Minor revision:

The authors introduce NMD pipeline to provide a potential mechanistic insight of how altered splicing events can lead to NMD in sepsis via a novel computational pipeline. Based on the RNA-Seq data of whole blood from 49 critically ill patients in the ICU and downstream analysis, it's demonstrated that the rate of NMD is higher in sepsis and deceased groups compared to control and survived groups, which signify purposeful downregulation of transcripts by AS-NMD or aberrant splicing due to altered physiology. And Grancalcin (GCA) was used to test the NMD pipeline to predict and evaluate the effect of NMD on protein abundance. NMD pipeline can potentially help discover protein targets previously unknown due to DGE studies alone. The NMD pipeline not only help in understanding the pathophysiological processes of sepsis but also offer new directions for its diagnosis and treatment. The manuscript can be improved in some aspects before accepted:

(1) Alternative splicing (AS) introduced PTC into RNA. The data analysis results showed that the sepsis and deceased group had a higher NMD rate. However, compared with the control group, there was no significant difference in the number of PTC. This seems to be contradictory to the significant difference in the NMD rate. Maybe it be caused by the difference in the location of PTC, please provide a reasonable explanation for this.

Thank you for your excellent point. We have included a potential explanation to Line 272:

“Thus, NMD pipeline not only highlights AS-NMD interaction, but also identifies potential protein targets previously unknown in sepsis. The fact that the number of PTCs was not significantly different in control vs. sepsis and survived vs. deceased is consistent with literature that shows NMD machinery is also sensitive to the location of PTC, such as downstream of last exon junction complex, near the stop codon, and long exons with far distances between the PTC and the stop codon.”

(2) Some genes of the sepsis and deceased group were less or more frequently spliced. Whether the sequences of these genes can be analyzed to find the potential specificity or commonality among them.

Thank you again for your insight. Unfortunately, we are unable to match the exact sequence of the genes and transcripts from their splicing events. The reason is that the 150 base pair RNA sequencing reads are processed by Whippet software to yield splicing event data but the software does not reveal which splicing event traces back to which exact sequence. However, a potential future direction would be to utilize an alternative software that does yield exact isoform sequences to enable such study. Please see Line 307 for revision.

Reviewer 2

In this manuscript, the authors analyze RNA-Seq data from sepsis and control human patients to determine whether altered RNA processing under these conditions generates putative Nonsense-Mediated Decay (NMD) substrates. While there are some potentially interesting observations-particularly regarding grancalcin-the connection to NMD remains unclear.

Overall I believe that there are interesting findings in their data but that the authors have to seriously reconsider how they analyze it and how they present their results. In light of this, I am not in favor of publication.

Major Concerns:

1- Clarity and Writing Quality:

The manuscript is poorly written and often difficult to understand. It contains excessive abbreviations-some of which are never defined (e.g., AF and AL in Figure 1D, H)-and many sections are nearly incomprehensible due to ambiguous phrasing and insufficient context (see "minor concerns" below for some examples).

We have added the definitions for AF and AL to Figure 1D, H, Line 509:

“(D) Proportion of each subtype out of all splicing events in control vs. sepsis groups then categorized into “Splicing” and “Transcription-related” groups. AF refers to alternative first exon and AL refers to alternative last exon.”

The rest of the abbreviations are defined in the text. Please see below under minor concerns section for the rest of our revisions.

2- Patient Group Composition:

The patient groups differ substantially in age (43 vs. 63 years) and gender (33% vs. 55% male) which may confound the results obtained.

We have added the potential confounding of age difference between control and sepsis group to Line 290:

“Some limitations include the potential confounding effect of the age difference between control and sepsis groups and non-NMD decay processes that could have influenced the transcript and protein levels. Our study does not establish definitive causality between AS-NMD machinery and gene expression in sepsis.”

The proportion of male and female in control versus sepsis group was not statistically significant as indicated in Table 1 (33.3% vs. 55.8%, $p=0.55$).

3- Magnitude of Alternative Splicing Events:

The manuscript never addresses the extent to which alternative RNA processing events differ between groups. It is possible that a particular exon is skipped 1% vs 5% of the time in sepsis vs control patients. This "statistically different" event is interesting, but very different in nature from the situation where a particular exon is skipped 1% vs. 95% in the two populations. We are never given a sense whether these "statistically different" events are small or large in magnitude.

The magnitude of splicing event difference between groups is measured by probability and delta percent-spliced-in (DeltaPsi) metric per Whippet software documentation as cited in

Methods section. Probability is a Bayesian estimate of a given splicing event differentially spliced between 2 groups and DeltaPsi is the percent difference in splicing between 2 groups, as your examples suggest.

We performed an additional analysis to calculate the magnitude of splicing events with appropriate contextualization using Anderson-Darling normality test, identifying the median of the absolute values of individual delta psi in control and sepsis groups, then using Wilcoxon rank-sum test. Median splicing percentages of control group were 1.98% and of sepsis group were 40.4% ($p < 0.0001$), demonstrating that sepsis group had statistically higher magnitude of splicing events compared to control group. Please see Line 97 for revision.

We initially did not include these metrics since our question focused on binary statistically significant differential splicing between control and sepsis as defined by Whippet software (probability > 0.9 , $|\Delta\Psi| > 0.1$). Our goal was to predict the rate of NMD from splicing events significantly different between control and sepsis, thus we followed Whippet documentation for this approach. Regardless, we have added the specific metrics for clarification. We have included this explanation to Line 139 of Supplementary Materials.

4- Interpretation of NMD-Linked Events:

The claim that exon skipping promotes NMD in $>66\%$ of cases implies that skipped exons are not frame-preserving at a higher rate than chance. This is contrary to established literature. Prior studies (e.g., PMID: 17719121, 39223315) show that at least 33% of cassette exons are frame-preserving. This goes up in conserved skipped exons. Thus AT MOST we expect that exon skipping should produce an NMD substrate 66% of the time. The analysis appears inconsistent with this and raises concerns about the validity of the NMD predictions. Similar issues exist for other types of alternative splicing documented by the authors.

We would like to emphasize that our methodology and the two papers cited above are fundamentally different, thus the results will imply discrepant findings. In addition, the major hypothesis of our work is that altered physiology of critical illness as seen in sepsis is associated with changes in alternative splicing. The two papers utilized human and mouse transcripts from Alternative Splicing Database (PMID: 17719121) and lymphoblastoid cell lines derived from 40-86 Yoruba individuals (PMID: 39223315) to yield their exon skipping results. We have prospectively obtained whole blood samples from critically ill patients with and without sepsis, perform RNA sequencing, then analyzed the raw data with Whippet to yield splicing events data. Our previous work has shown that critical illness affects splicing events to modify downstream protein translation (1, 2), specifically how physiologic derangement leads to a change in exon skipping events that render transmembrane proteins to become soluble. Given sepsis is marked by physiologic derangement, it is not inconsistent, but actually expected, that the exon skipping results from our cohort of critically ill patients had different exon skipping frequency compared to non-critically ill humans, mice, and cell lines from the two studies. Although causality can't be established with the current data, there is work showing clinically relevant physiologic derangements (pH, temperature, hypoxia) impact alternative splicing. We have included this revision to Line 245:

“While these data provide helpful reference, our study has utilized prospectively obtained whole blood samples from critically ill patients with and without sepsis, performed RNA sequencing, then analyzed the raw data with Whippet to yield splicing events data. The two prior studies utilized human and mouse transcripts from Alternative Splicing Database and lymphoblastoid cell lines derived from 40-86 Yoruba individuals to yield exon skipping results, which are implicated for healthy human and mouse splicing data. In addition, the major hypothesis of our work is that altered physiology of critical illness as seen in sepsis is associated with changes in alternative splicing, as our previous work has shown that critical illness affects splicing events to modify downstream protein translation, specifically how physiologic derangement leads to a change in exon skipping events that render transmembrane proteins to become soluble. Given sepsis is marked by physiologic derangement, it is not inconsistent, but expected that the exon skipping results from our cohort of critically ill patients had different exon skipping frequency compared to non-critically ill humans, mice, and cell lines from the two studies. Although causality can’t be established with the current data, our results can supplement prior studies by reporting higher percentage of exon skipping events are predicted to induce NMD in critically ill control group, which indicates critical illness can affect splicing percentages.”

- 1. Monaghan SF, Chung CS, Chen Y, Lomas-Neira J, Fairbrother WG, Heffernan DS, et al. Soluble programmed cell death receptor-1 (sPD-1): a potential biomarker with anti-inflammatory properties in human and experimental acute respiratory distress syndrome (ARDS). *J Transl Med.* 2016;14(1):312.**
- 2. Monaghan SF, Banerjee D, Chung CS, Lomas-Neira J, Cygan KJ, Rhine CL, et al. Changes in the process of alternative RNA splicing results in soluble B and T lymphocyte attenuator with biological and clinical implications in critical illness. *Mol Med.* 2018;24(1):32.**

5- Speculative Interpretation of NMD Rates:

In the discussion, the authors suggest that increased NMD in sepsis is due to aberrant splicing, which is highly plausible. However, as discussed above, the observed NMD levels exceed those expected from random or non-functional RNA generation. This discrepancy is never addressed and again suggests a possible flaw in the NMD prediction pipeline.

Please see above comment #4 and in addition:

We would like to clarify the following points. First, our NMD pipeline is built to use splicing event outputs from Whippet software as inputs to predict the rate of NMD. Whippet (3) operates on a peer-reviewed and established computational pipeline that yields splicing events from RNA-Seq data. In addition, we use statistical significance threshold recommended by Whippet documentation (4) to analyze only the splicing events more likely to be significantly different between control and sepsis groups. Third, coding algorithm in our pipeline was written based on one of the conventional ways that NMD machinery is triggered – introduction of premature stop codon from a frameshift splicing event at 50-55 base pairs upstream of final exon junction (5). Thus, we believe that our pipeline is correctly identifying splicing events since it has adhered to evidence-based, validated methods from input selection to coding implementation.

Regarding further experimental validations of our NMD pipeline, we have validated our computational work with the ELISA validation of the protein Grancalcin (GCA). This work utilized clinical samples paired with samples taken from patients for RNA sequencing. We are not able to obtain more blood from these patients as some patients were recruited as early as 2021. In addition, we do not think animal model experiments are appropriate for sepsis work and this is consistent with NIGMS (funding organization of the R35) policies specific to sepsis (<https://www.nigms.nih.gov/sites/nigms/files/migrated/nagmsc-working-group-sepsis-report.pdf>) with current plans to expand this across the entire NIH. Our lab has focused on human samples, RNA sequencing, and computational methods with focused experimental validations such as ELISA as stated earlier and we do not believe an animal model would translate this work. Thus, we acknowledge our limitations and will edit the manuscript if acceptable to remove speculative language.

Please see Line 295 for revisions:

“However, we do provide a potential mechanistic insight of how altered splicing events can lead to NMD in sepsis via a novel computational pipeline using critically ill patient samples to augment translational impact and with adherence to evidence-based guidelines. First, our NMD pipeline is built to use splicing event outputs from Whippet software as inputs to predict the rate of NMD. Whippet operates on a peer-reviewed and established computational pipeline that yields splicing events from RNA-Seq data. In addition, we use statistical significance threshold recommended by Whippet documentation (30) to analyze only the splicing events more likely to be significantly different between control and sepsis groups. Third, coding algorithm in our pipeline was written based on one of the conventional ways that NMD machinery is triggered – introduction of premature stop codon from a frameshift splicing event at 50-55 base pairs upstream of final exon junction. Thus, despite the above limitations, our results have undergone validations with ELISA and have adhered to evidence-based, validated methods from input selection to coding implementation.”

3. Sterne-Weiler T, Weatheritt RJ, Best AJ, Ha KCH, and Blencowe BJ. Efficient and Accurate Quantitative Profiling of Alternative Splicing Patterns of Any Complexity on a Laptop. *Mol Cell*. 2018;72(1):187-200 e6.
4. T. Sterne-Weiler NA, G. Massman. Whippet. <https://github.com/timbitz/Whippet.jl>.
5. Lareau LF, Brooks AN, Soergel DA, Meng Q, and Brenner SE. The coupling of alternative splicing and nonsense-mediated mRNA decay. *Adv Exp Med Biol*. 2007;623:190-211.

Minor Comments:

1- Figure 1B-C lacks clear labeling. It's unclear whether the X-axis represents sepsis/control ratios or the reverse.

Amends have been made to Lines 475, 478, 485, 487 of Figure 1B-C and Figure 1F-G captions:

“X-axis (Log₂ FC) represents Log₂ of sepsis/control.”

“X-axis (Delta Psi) represents percentage of splicing in sepsis subtracted by percentage of splicing in control.”

“X-axis (Log2 FC) represents Log2 of deceased/survived.”

“X-axis (Delta Psi) represents percentage of splicing in deceased subtracted by percentage of splicing in survived.”

2- The way the text is written is sometimes ambiguous and hard to parse. Here are several examples:

Line 71. "The sepsis group was older than the controls (vs, $p=0.049$)" - logically the numbers should follow the order of the text and should be "63.2 {plus minus} 14.9 years vs, 43.8 {plus minus} 18 years". I know that we can infer that the opposite order is implied, but this cannot be done for the following statements (e.g., which group, sepsis or control, has 33% males?).

For consistency, the entire paragraph was written to follow the order of control vs. sepsis and survived vs. deceased. For further clarity, we have made amends to Line 71 to reflect such order:

“The control group was younger than sepsis group (43.8 ± 18 years vs. 63.2 ± 14.9 years, $p=0.049$) but did not have significantly different percentages of male (33.3% vs. 55.8%, $p=0.55$) or non-Caucasians (50% vs. 27.9%, $p=0.53$).”

Line 91 "Of the 17,043 genes differentially expressed between the two groups ..." This is not correct and very confusing. It should read "Of the 17,043 genes analyzed in the two groups ... "

Revision has been made to Line 91: “Of the 17,043 genes analyzed in the two groups, 1,349 genes (7.9%) were significantly differentially expressed with 1,325 upregulated (98.2%) and 24 (1.8%) downregulated in sepsis showing that more genes analyzed were highly expressed in sepsis (Figure 1B, S1).”

Line 94 "There were 220,779 events differentially spliced in control versus sepsis ..." I presume that you mean that there were 220,779 RNA processing/splicing events analyzed in control and sepsis samples, since only 1% were significantly different. The rest of the paragraph, and the entire next section, were extremely hard to parse. When the author express a fact, such as "alternative transcription events represented 48.1%" it is imperative that they express what they exactly is the reference point (i.e. 48.1% of what group?).

Revision has been made to Line 94 and 103:

“There were 220,779 splicing events analyzed in control versus sepsis, with 2,158 splicing events (1%) significantly differentially frequent. Of these, 1,014 events (47%) were more frequent in sepsis and 1,144 events (53%) less frequent, highlighting that splicing events were more evenly split in contrast to DGE results (Figure 1C, S2).”

“The results showed that alternative transcription events represented 48.1% and 55% of all splicing events in each group, with transcription start (TS) and end (TE) constituting the highest percentages (94.5%, 95.6%), whereas splicing events were 51.9% and 45% in

each group with exon skipping events (ES) constituting the highest percentages (76.3%, 44.7%) (Figure 1D, Table S1)."

3- The section starting on page 170 analyzes putative functional consequences of alterations in RNA processing that result in non-NMD substrates. The paucity of hits (13 and 11 genes) makes this unlikely and Figure 2H-1 is meaningless. The authors claim "the NMD pipeline can identify proteins with known sepsis-relevant or essential cellular functions ... not previously known in sepsis" But the changes observed may not be relevant to sepsis and may simply be serendipitous. Leaving aside the paucity of affected genes, how do the authors know that these are relevant to sepsis? Correlation does not mean causation. This is akin to finding that one difference between parked and moving cars is that the former have parking tickets and concluding that the parking tickets caused the cars to become parked.

The purpose of Figures 2H-I is to show that our NMD pipeline can help identify potential proteins not previously studied in sepsis. The rationale is that genes/transcripts not predicted to undergo NMD based on our pipeline may be important to study in sepsis since they are part of the minority of genes less likely to get decayed by NMD. Thus, the number of hits (13 and 11 genes) does not undermine our rationale as we purposefully selected the most statistically significant hits based on GO terms as indicated in Line 174. Revisions have been made to Line 173 for further clarity:

"We then examined whether splicing events not predicted to induce NMD, thus preserving certain transcripts, could identify proteins with potentially novel roles in sepsis and mortality. The rationale was that genes not predicted to undergo NMD based on our pipeline may be important to study in sepsis since they are part of the minority of genes less likely to be decayed."

We do acknowledge our findings are correlative, not causative. We indicate this limitation in Line 290 as above.

Our statement regarding "sepsis-relevant" was specifically referring to known pathways such as NF-kB. To prevent confusion, we have removed that phrase in Line 185:

"Thus, NMD-F from the NMD pipeline can identify proteins with essential cellular functions (i.e. inflammation, nucleic acid and cell metabolism) and potential novel roles (i.e. response to alcohol, UV) not previously known in sepsis."

4- Line 187 "Grancalcin (GCA) was chosen since it was the only gene with significant differential splicing events in both sepsis and deceased groups, each with 2,656 and 866 events," I do not know what these event counts mean. Are these individual reads? How do these events in the aggregate differ between the groups?

Please refer to Figure 3 captions for explanation: "control vs sepsis (total 2,656 significant differential splicing events) and survived vs deceased (total 866 significant differential splicing events)". For clarity, we have removed these numbers from the main text, Line 191:

“Grancalcin (GCA) was chosen since it was the only gene with significant differential splicing events in both sepsis and deceased groups and with one of the highest RNA-Seq read counts overall in both groups (Figure 3A, Supplementary Text).”

5- Figure 3I, the two trends are identical except for a few outliers. This should be eliminated.

As legends suggest, these two graphs refer to 2 different analyses: Fig. 3E for control vs. sepsis and Fig. 3I for survived vs. deceased. We believe it is important to show both graphs since they suggest findings from sepsis and mortality analyses:

“(E) Graph showing the correlation data between ELISA concentrations in ng/mL and RNA-Seq read counts of GCA in control vs sepsis.”

“(I) Graph showing the correlation data between ELISA concentrations in ng/mL and RNA-Seq read counts of GCA in survived vs deceased.”

Reviewer 3

My comments are as follows that I would like the authors to address:

1. The study claimed about the development of the pipeline/method for detecting and assessing nonsense-mediated mRNA decay (NMD) but it is unclear if validation of the performance was performed.

To be clear the method is not detailed in the main text. There might be something in the supplemental doc that I missed

Thank you for your point. We have validated our NMD computational pipeline with ELISA results as shown in Figure 3. In addition, we have adhered to evidence-based, validated methods from input selection to coding implementation in building our pipeline. First, our NMD pipeline is built to use splicing event outputs from Whippet software as inputs to predict the rate of NMD. Whippet operates on a peer-reviewed and established computational pipeline that yields splicing events from RNA-Seq data. In addition, we use statistical significance threshold recommended by Whippet documentation to analyze only the splicing events more likely to be significantly different between control and sepsis groups. Third, coding algorithm in our pipeline was written based on one of the conventional ways that NMD machinery is triggered – introduction of premature stop codon from a frameshift splicing event at 50-55 base pairs upstream of final exon junction. External validation with a separate cohort has not been done and is one of our future directions.

Please see Line 293 for revisions:

“For instance, the AS and NMD interplay in GCA has not been validated experimentally on a molecular level due to limitations in obtaining further blood samples and available laboratory resources. However, we do provide a potential mechanistic insight of how altered splicing events can lead to NMD in sepsis via a novel computational pipeline using critically ill patient samples to augment translational impact and with adherence to evidence-based guidelines. First, our NMD pipeline is built to use splicing event outputs from Whippet software as inputs to predict the rate of NMD. Whippet (23) operates on a peer-reviewed and

established computational pipeline that yields splicing events from RNA-Seq data. In addition, we use statistical significance threshold recommended by Whippet documentation (30) to analyze only the splicing events more likely to be significantly different between control and sepsis groups. Third, coding algorithm in our pipeline was written based on one of the conventional ways that NMD machinery is triggered – introduction of premature stop codon from a frameshift splicing event at 50-55 base pairs upstream of final exon junction (4). Thus, despite the above limitations, our results have undergone validations with ELISA and have adhered to evidence-based, validated methods from input selection to coding implementation.”

The Method section is placed after the Discussion section. The supplementary figures, tables, and text are uploaded separately and include more detailed descriptions of our pipeline methodology.

2. It is unclear if detection of AS and NMD of genes such as GCA were experimentally validated.

We did experimentally validate the potential role of AS-NMD by measuring protein concentration via ELISA as shown in Figure 3. However, we did not study AS and NMD of GCA on a molecular level due to the following limitations:

Regarding further experimental validations of our NMD pipeline, we have validated our computational work with the ELISA validation of the protein Grancalcin (GCA). This work utilized clinical samples paired with samples taken from patients for RNA sequencing. We are not able to obtain more blood from these patients as some patients were recruited as early as 2021. In addition, we do not think animal model experiments are appropriate for sepsis work and this is consistent with NIGMS (funding organization of the R35) policies specific to sepsis (<https://www.nigms.nih.gov/sites/nigms/files/migrated/nagmsc-working-group-sepsis-report.pdf>) with current plans to expand this across the entire NIH. Our lab has focused on human samples, RNA sequencing, and computational methods with focused experimental validations such as ELISA as stated earlier and we do not believe an animal model would translate this work. Thus, we acknowledge our limitations and will edit the manuscript if acceptable to remove speculative language.

Please see Line 293 for revisions.

3. The control group seems to have small sample size (n=6) in comparison.

We acknowledge that our control group has a small sample in comparison to sepsis group. However, we believe the depth of our RNA-Seq data yielding at least 100 million reads per sample provided sufficient data to investigate gene expression, splicing events, and NMD prediction. The reason is that though sample size was 6, total number of transcripts included in the analysis amounted to more than 600 million in the control group. We supplemented this information in Line 310 for clarity:

“While the control group had a relatively small sample size compared to sepsis group, from our sample size of around 50 patients, the depth of our RNA-Seq data covering at least 100

million RNA reads from each patient provided sufficient data points for our current investigation of predicting the rate of NMD from splicing events”

4. 'While DGE data showed upregulation of genes in sepsis and downregulation of genes in deceased patients, AS data showed transcripts were differentially spliced in nearly equivalent levels in both groups. This suggests AS may modulate downstream effects of over- or under-translation of proteins to maintain cellular homeostasis in diseased states (24). ' Its unclear as written/data illustrated that it has that implication.

We have changed the language of the sentence reflecting your comment. Please see Line 234:

“Given AS has been suggested to maintain cellular homeostasis in diseased states, the distribution of splicing data unique from DGE indicates the potential importance of including AS in gene expression studies.”

Overall, the work is novel and impressive.

Thank you for your acknowledgement.

August 25, 2025

RE: Life Science Alliance Manuscript #LSA-2025-03380-TR-A

Dr. Sean F Monaghan
Rhode Island Hospital
Surgery
593 Eddy St
Middle House 211
Providence, RI 02903

Dear Dr. Monaghan,

Thank you for submitting your revised manuscript entitled "Predicting Nonsense-mediated mRNA Decay from Splicing Events in Sepsis using RNA-Sequencing Data". This revised manuscript was evaluated by the original reviewers, whose reports are below.

As you will see, Reviewer 2 remains concerned that framing and overall significance of this study remains poorly conveyed, and also notes several points in the text that must be improved. Although the description of splicing events and NMD in sepsis patients provided here is valuable, the text should frame these findings clearly and without overstating the evidence provided. We would be happy to publish your paper in Life Science Alliance pending resolution of all remaining points by Reviewer 2 as well as final revisions necessary to meet our formatting guidelines.

- Please upload your main manuscript text as an editable doc file.
- Please upload all figure files as individual ones, including the supplementary figure files; all figure legends should only appear in the main manuscript file.
- Please add a Summary Blurb/Alternate Abstract in our system.
- Please add the X and Bluesky handles of your host institute/organization, as well as your own and/or one of the authors, in our system.
- The "Data Availability" section should be placed after the Materials & Methods section. Please consult our guidelines at <https://www.life-science-alliance.org/manuscript-prep#format>.
- The contributions selected for Brandon E Armstead, Alfred Ayala, William G. Fairbrother, Kwesi K Lillard, and Gerard J Nau do not qualify them for authorship. Please either update the contributions in our system and in the Author Contributions section of the manuscript, or let us know if the authors need to be removed (and potentially added to the acknowledgment section).
- Please add your main figure, supplementary figure, and table legends to the main manuscript text after the references section.
- Please add the Supplementary Materials and Methods text to the main Methods section.
- Please upload your Tables in editable .doc or Excel format.
- Please add callouts for Figures S5 and S6 to your main manuscript text.

LSA now encourages authors to provide a 30-60 second video where the study is briefly explained. We will use these videos on social media to promote the published paper and the presenting author (for examples, see <https://docs.google.com/document/d/1-UWCfbE4pGcDdcgzcmiuJl2XMBJnxKYeqRvLLrLS08s/edit?usp=sharing>). Corresponding or first-authors are welcome to submit the video. Please submit only one video per manuscript. The video can be emailed to contact@life-science-alliance.org

A. FINAL FILES:

- An editable version of the final text (.DOC or .DOCX) is needed for copyediting (no PDFs).

B. MANUSCRIPT ORGANIZATION AND FORMATTING:

Thank you for your attention to these final processing requirements. Please revise and format the manuscript and upload materials as soon as you are able.

Sincerely,

Reviewer #1 (Comments to the Authors (Required)):

The author has addressed my questions very well, and after further refining some descriptions, I suggest that this manuscript can be accepted.

Reviewer #2 (Comments to the Authors (Required)):

While there are some potentially interesting observations, the connection between sepsis and NMD remains unclear. Are NMD changes relevant/causative/treatable? This is never addressed. It is unclear whether the results are informative and this greatly dampens my enthusiasm.

Beyond this major criticism, the manuscript has improved somewhat, but there are still major deficiencies.

1) There are still problems with the text. Although some have been fixed, others have not. Many articles are missing, and nouns that should be pluralized are often written in singular. Infuriatingly, I pointed to a subset of problems in my last review, and although the exact examples that I pointed out were corrected, other problematic sections, some with the exact same errors as the examples that I pointed out previously, persist. Some examples:

"Of the 16,837 genes differentially expressed ..." and "There were 233,753 differential splicing events ..."

2) In many cases the authors still rattle off percentages, but it is not always clear what the denominator is or what they are referring to. For example "Of note, median splicing percentages of control group were 1.98% and of sepsis group were 40.4% ($p < 0.0001$)" The authors need to place themselves in the shoes of the readers, and ask themselves, is this crystal clear?

3) Lines 185-188. The text implies that that identified genes/proteins with changes in processing are important for sepsis - this should be changed. We do not know that they play a role in sepsis. It could simply be that they just happened to be altered.

4) Increases in NMD-substrate levels may be due to changes in RNA processing (as the authors seem to imply) or due to a decrease in NMD activity (leading to an increase in NMD targets which have become stabilized). This caveat should be acknowledged.

Reviewer #3 (Comments to the Authors (Required)):

Critiques addressed.

Editorial

As you will see, Reviewer 2 remains concerned that framing and overall significance of this study remains poorly conveyed, and also notes several points in the text that must be improved. Although the description of splicing events and NMD in sepsis patients provided here is valuable, the text should frame these findings clearly and without overstating the evidence provided. We would be happy to publish your paper in Life Science Alliance pending resolution of all remaining points by Reviewer 2 as well as final revisions necessary to meet our formatting guidelines.

Thank you very much for your continued interest and support of our manuscript. All remaining points by Reviewer 2 have been addressed as shown below and reflected in the final manuscript. We also changed the language of our findings to improve clarity and avoid overstatement.

- Please upload your main manuscript text as an editable doc file.
- Please upload all figure files as individual ones, including the supplementary figure files; all figure legends should only appear in the main manuscript file.
- Please add a Summary Blurb/Alternate Abstract in our system.
- Please add the X and Bluesky handles of your host institute/organization, as well as your own and/or one of the authors, in our system.
- The "Data Availability" section should be placed after the Materials & Methods section. Please consult our guidelines at <https://www.life-science-alliance.org/manuscript-prep#format>.
- The contributions selected for Brandon E Armstead, Alfred Ayala, William G. Fairbrother, Kwesi K Lillard, and Gerard J Nau do not qualify them for authorship. Please either update the contributions in our system and in the Author Contributions section of the manuscript, or let us know if the authors need to be removed (and potentially added to the acknowledgment section).
- Please add your main figure, supplementary figure, and table legends to the main manuscript text after the references section.
- Please add the Supplementary Materials and Methods text to the main Methods section.
- Please upload your Tables in editable .doc or Excel format.
- Please add callouts for Figures S5 and S6 to your main manuscript text.

All of these points have been addressed.

Reviewers

Reviewer #1 (Comments to the Authors (Required)):

The author has addressed my questions very well, and after further refining some descriptions, I suggest that this manuscript can be accepted.

Thank you very much for your support and time.

Reviewer #2 (Comments to the Authors (Required)):

While there are some potentially interesting observations, the connection between sepsis and NMD remains unclear. Are NMD changes relevant/causative/treatable? This is never addressed. It is unclear whether the results are informative and this greatly dampens my enthusiasm.

Relevance: NMD changes are relevant in sepsis because our prior studies have shown that the altered physiologic state in critical illnesses such as sepsis changes the type of splicing events that transcripts undergo, thereby altering the protein translation. NMD is one of the downstream effects of a change in splicing events that could occur in sepsis. In addition, AS and NMD are part of a known coupling machinery that modulates gene expression. Thus, studying NMD in the context of AS and sepsis is relevant both clinically and biologically. Please see Lines 60-61, 251-261, 263-274, 297-300, and 327-339.

Causative: Please see Line 305-310 for the statement regarding causality: “Our study does not establish definitive causality between AS-NMD machinery and gene expression in sepsis. For instance, the AS and NMD interplay in GCA has not been validated experimentally on a molecular level due to limitations in obtaining further blood samples and availability in laboratory resources. However, we do provide a potential mechanistic insight of how altered splicing events can lead to NMD in sepsis via a novel computational pipeline developed based on the evidence-based guidelines.”

Treatable: NMD itself is not a treatable target- it is an essential, highly conserved cellular machinery that cannot be targeted to silence or modify given the downstream ramifications on the eukaryotic cellular machinery. However, its mechanistic coupling with AS can be studied in conjunction to study the gene expression pathway to better understand sepsis pathogenesis and identify protein targets. Please see Line 274-277.

To emphasize and further clarify, we have re-summarized all the above points at the end of our manuscript in Line 327-339:

“Overall, this study demonstrates that the NMD pipeline can predict NMD from splicing events. While NMD changes alone are not causative of sepsis or

directly treatable as targets, our pipeline enables the study of AS-NMD and their potential interplay in sepsis and build upon the previous literature on the role of splicing in critical illness. In the sepsis and deceased groups, we report higher rates of NMD, either from aberrant splicing requiring more frequent NMD or purposeful downregulation of certain genes. We also propose that the NMD pipeline can be a component of gene expression studies, both in sepsis and other fields, to characterize post-transcriptomic gene products at levels detected under homeostasis and pathological conditions. Studying AS-NMD in conjunction with differential gene expression studies can capture the nuances of complex gene expression pathway in sepsis, for which DGE analysis alone may not suffice. Further, investigating the role of AS-NMD in altered physiological states can enhance current understanding of the role of splicing in critical illness and uncover potential proteins associated with sepsis.”

Beyond this major criticism, the manuscript has improved somewhat, but there are still major deficiencies.

1) There are still problems with the text. Although some have been fixed, others have not. Many articles are missing, and nouns that should be pluralized are often written in singular. Infuriatingly, I pointed to a subset of problems in my last review, and although the exact examples that I pointed out were corrected, other problematic sections, some with the exact same errors as the examples that I pointed out previously, persist. Some examples:

"Of the 16,837 genes differentially expressed ... " and "There were 233,753 differential splicing events ..."

We have reviewed the entire manuscript again and rectified the listed issues: missing articles, singular vs. plural nouns, and potentially confusing languages.

2) In many cases the authors still rattle off percentages, but it is not always clear what the denominator is or what they are referring to. For example "Of note, median splicing percentages of control group were 1.98% and of sepsis group were 40.4% ($p < 0.0001$)" The authors need to place themselves in the shoes of the readers, and ask themselves, is this crystal clear?

We have audited all the percentage values again and confirmed that they have the proper descriptors to explain the denominators and numerators. For instance,

“Of note, the median percent-spliced in (psi) value in Whippet – the proportion of reads that include splicing events in the final transcript sequence across samples– of the control group were 1.98% and of the sepsis group were 40.4% ($p < 0.0001$), demonstrating that the sepsis group had a statistically higher magnitude of splicing events compared to the control group.” (Line 99-103)

3) Lines 185-188. The text implies that that identified genes/proteins with changes in processing are important for sepsis - this should be changed. We do not know that they play a role in sepsis. It could simply be that they just happened to be altered.

We have fixed the languages to avoid overstating the significance. Examples are:

Line 188-189: Thus, NMD-F from the NMD pipeline can identify proteins with essential cellular functions such as inflammation and nucleic acid and cell metabolism.

Line 268-271: a list of genes identified as NMD-F (Fig. 2H, 2I) have noteworthy involvement in nucleic acid and cell metabolism, signal transduction, inflammation, and response to stressor as evidenced by GO enrichment analysis.

Line 272-274: Thus, the NMD pipeline not only highlights the AS-NMD interaction, but also identifies some potential proteins with roles in inflammation and essential cellular metabolism.

4) Increases in NMD-substrate levels may be due to changes in RNA processing (as the authors seem to imply) or due to a decrease in NMD activity (leading to an increase in NMD targets which have become stabilized). This caveat should be acknowledged.

We have acknowledged this caveat in Line 303-305:

“In addition, an increase in NMD-substrate levels may also be attributed to a decrease in NMD activity – thus leading to an increase in NMD targets – which the scope of our study does not include but can be a plausible mechanism.”

Reviewer #3 (Comments to the Authors (Required)):

Critiques addressed.

Thank you very much for your support and comments.

September 15, 2025

RE: Life Science Alliance Manuscript #LSA-2025-03380-TRR

Dr. Sean F Monaghan
Rhode Island Hospital
Surgery
593 Eddy St
Middle House 211
Providence, RI 02903

Dear Dr. Monaghan,

Thank you for submitting your Resource entitled "Predicting Nonsense-mediated mRNA Decay from Splicing Events in Sepsis using RNA-Sequencing Data". It is a pleasure to let you know that your manuscript is now accepted for publication in Life Science Alliance. Congratulations on this interesting work.

DISTRIBUTION OF MATERIALS:

Again, congratulations on a very nice paper. I hope you found the review process to be constructive and are pleased with how the manuscript was handled editorially. We look forward to future exciting submissions from your lab.

Sincerely,
